# Autistic adults' experiences of accessing and receiving mental health care and their priorities for improvements: A qualitative study

Frederick Taylor[1], Nafiso Ahmed[1]*, Tamara Pemovska[1,2], Farisa Dar[1], Brynmor Lloyd-Evans[1], Sonia Johnson[1,3]

**1** Division of Psychiatry, University College London, London, United Kingdom, **2** Centre for Evidence and Implementation, London, United Kingdom, **3** North London NHS Foundation Trust, London, United Kingdom

* nafiso.ahmed@ucl.ac.uk

## Abstract

Autistic adults are disproportionately affected by mental health conditions yet face significant barriers in accessing support and receiving suitable care. To understand this disparity better, using qualitative methods, we investigated the experiences of autistic adults in accessing and receiving mental health care and their priorities for improvement. Thirteen autistic adults with experience of mental health services in England were purposively selected to participate in semi-structured interviews. Reflexive thematic analysis principles guided the analysis of interview transcripts. Participants felt that an understanding of autism was key for professionals to provide effective support, and reported this was often lacking, resulting in insufficient recognition for their autism-specific needs, feelings of neglect and inadequate treatment. Inconsistent services, including varying appointments and unfamiliar professionals exacerbated anxiety and hindered treatment benefits. A recurring theme of validation emerged as participants expressed a shared sense of not being believed or taken seriously within the context of mental health support. In terms of priorities for improving services, adaptability was a central focus, while better understanding and addressing the specific needs of autistic people was considered crucial. Participants - females in particular – emphasised the need for professionals to adopt more effective communication strategies as they felt that poor communication and misunderstandings often delayed autism diagnoses. To enhance mental health support for autistic adults and mitigate negative outcomes, increasing mental health professionals' understanding of autism is vital. Further research is needed to understand the manifestations and risk factors of mental health conditions in autistic adults. Participants stressed that autistic people need to be directly involved in driving these priorities and guiding the enhancement and adaptation of mental health care to meet their needs.

**Data availability statement:** The data underlying the results of this study cannot be shared due to ethical restrictions on sharing a de-identified dataset. In compliance with the protocol approved by the University College London Research Ethics Committee (Ref. 25209/001), individual transcripts cannot be publicly shared, as participants did not consent to their disclosure. Public access would also compromise patient confidentiality. For data access inquiries, please contact the corresponding author (NA) or the following independent institutional body: UCL Research Ethics Committee Email: ethics@ucl.ac.uk.

**Funding:** This study was conducted as the dissertation project for the first author's MSc studies at University College London. Reimbursement to participants and lived experience researchers, as well as transcription costs, were supported by funds from the UCL course budget available to all projects within the MSc cohort. UCL had no role in the study design, data collection and analysis, decision to publish, or preparation of the manuscript.

**Competing interests:** I have read the journal's policy and the authors of this manuscript have the following competing interests: NA, BLE and SJ currently work at the same university, UCL, as the editor chief of PLOS Mental Health, RB. All other authors declare no competing interests.

## Introduction

Autistic adults experience a significantly higher incidence of mental ill-health compared to the general population [1], with markedly elevated rates of anxiety disorders (20% compared to 7.3%), depressive disorders (11% compared to 4.7%), bipolar and related disorders (5% compared to 0.71%), schizophrenia (4% compared to 0.46%) and obsessive-compulsive disorder and related disorders (9% compared to 0.7%) [2]. Having a mental health condition is associated with a lower quality of life for autistic people [3] and an increased risk of suicidal thoughts and behaviours [4]. Despite this, accessing appropriate mental health services has been found to be challenging for this population; twice as many autistic people report that they have unmet mental health needs compared to non-autistic people [5] and it has been claimed that current support systems are 'not fit for purpose' [6]. Without effective treatment and support, autistic people's mental health issues are likely to compound and ultimately become increasingly costly and harmful not only for the individuals involved but also the services that exist to support them.

Practitioners often lack the training and support needed to understand the intersection of autism and mental health and associated needs [7,8] As a result, misdiagnosis is common [9] with individuals often diagnosed with a mental illness before their autism is recognised [10]. Stereotypical views exacerbate this, particularly for autistic women and girls who may mask their traits effectively [11] or for those without intellectual disability who may face harmful assumptions about being 'high functioning' and able to cope [12]. Existing service frequently fail to accommodate the specific needs of autistic people, with reports of individuals being rejected as 'too complicated' or 'batted back and forth between agencies' [12], especially distressing given the reliance on routine and predictability to manage anxiety and sensory sensitivity [13] Qualitative studies have highlighted the urgent need for change in mental health support for young autistic adults in England [14] and while tailored support can produce positive outcomes [8], autistic individuals and their families report that services are rarely adapted [12] leading to autistic people 'falling through the gaps' [8].

There has been growing recognition of this issue from policymakers, with The World Health Organisation [15] highlighting the unmet needs of autistic adults as a public health concern worldwide, while in the UK, the NHS Long Term Plan [16] made it a priority to improve the mental health of autistic people over the next decade. In line with these efforts, the Health and Care Act 2022 introduced a statutory requirement that regulated service providers must ensure their staff receive learning disability and autism training appropriate to their role [17] While these initiatives and recent research such as reviews on adaptations to mental health care for autistic children [18] and adults [19] highlight efforts to improve access and quality of care, few of these approaches are well established or evidence-based, highlighting the need for further research and to incorporate the perspectives of autistic mental health service users themselves to establish priorities for improvement.

Although there is a growing body of qualitative research exploring the experiences of autistic adults receiving mental health support from their own perspective, many of these studies focus on just one area; these include mental health diagnoses

[9] specific interventions [20] and services related to a specific condition such as eating disorder [21]; other studies focus on one specific setting like acute inpatient psychiatric services [22] or transition between services [23]. In one of the few qualitative studies exploring autistic people's experiences of mental health services in general, an online survey was used rather than interviews [8]. Interviews offer a more flexible approach than surveys, particularly for autistic participants, as the interviewer can adapt to the participant's individual needs and communication styles and provide clarification if necessary. Interviews can also provide a more in-depth and organic approach in general, leading to richer data collection.

A key strength of the current study is its broader focus on autistic adults' experiences across general mental health services, enabling identification of common barriers and priorities for improvement that apply across diverse contexts and providing insights that are more widely applicable to policy and practice in adult mental health care.

## Aims

This study employed qualitative methods to ask autistic adults themselves the following overarching research questions:

1.  What are autistic adults' experiences of accessing and receiving mental health care?

2.  What are autistic adults' priorities for improving the mental health care they receive?

## Method

This study was carried out with the support of members of the National Institute for Health and Care Research (NIHR) Policy Research Unit in Mental Health team and complemented their programme of research on the mental health needs of autistic people [18,19]; it was also aided by members of the unit's Lived Experience Researcher group - a group of autistic adults with lived experience of mental health difficulties who are employed as paid researchers within the unit, and are regularly consulted regarding study questions, design, interpretation of findings and dissemination across studies. Two Lived Experience Researchers provided detailed written and verbal feedback on early drafts of the interview guide, suggesting rephrasing of some questions to make them clearer and more relevant to autistic service users and proposing additional prompts. They also reviewed emerging themes during analysis to ensure the interpretation remained grounded in autistic lived experience. The authors assert that all procedures contributing to this work comply with the ethical standards of the relevant national and institutional committees on human experimentation and with the Helsinki Declaration of 1975, as revised in 2013. All procedures were approved by the University College London Research Ethics Committee (Ref. 25209/001).

## Participants

We aimed to recruit between 12–15 participants who were over 18 years of age, identified as autistic (including those with a formal diagnosis and those who self-identified as autistic) and had experience of accessing and receiving mental health care in England for co-occurring mental health conditions This included any contact with specialist NHS mental health services such as community mental health teams, early intervention in psychosis services, home treatment teams, outpatient talking therapies or inpatient psychiatric care. The sampling strategy was purposive, aiming to achieve a diverse sample in terms of gender, ethnicity, age and types of mental health service used. Individuals with co-occurring intellectual disabilities were not specifically targeted in recruitment and were excluded as the study materials and interview format were designed for autistic adults without intellectual disability. The sample size of 12–15 participants is typical for qualitative research [24] to enable researchers to delve deeply into the diverse experiences and priorities of the participants.

Thirteen autistic adults with experience of accessing and receiving mental health care in England were recruited. The aim was not to achieve a statistically representative sample of participants and generalise the results to all autistic adults

with experience of mental health services in the England; but to assemble a diverse group of participants with regard to demographic characteristics and types of service used.

## Procedures

**Recruitment and consent.** The study was advertised on the National Survivor User Network's (NSUN) weekly bulletin to their network of over 5,000 members. This generated approximately forty expressions of interest. There were plans to recruit through other channels, but we received a very positive response from NSUN — a nationwide network that does not focus on any specific mental health condition or demographic group, making it a strong route to recruiting a diverse sample. Recruitment remained open while interviews were conducted, and demographic information (age, gender, ethnicity, sexual orientation, and types of mental health service use) was collected only after consent during the interviews themselves. Once the target sample size was reached and the consented sample demonstrated good demographic variability across these characteristics, providing a rich data set with a full range of perspectives, recruitment was closed. It was therefore decided to recruit solely from this group using a pragmatic approach, prioritising individuals who confirmed their eligibility through initial email discussions with the first author. No participants dropped out after interviews were arranged. Potential participants were invited to contact the author via email and were then sent the study participant information sheet. This sheet explained the purpose of the study (to explore autistic adults' experiences of mental health care in England and their priorities for service improvements), what participation would involve, the voluntary nature of participation, confidentiality procedures and possible emotional risks and available support. It also provided an overview of the main topics to be discussed. The full interview questions were not shared in advance to allow for a flexible, semi-structured conversation that could adapt to each participant's communication style and priorities.

Once participants had read the information sheet, online video calls were arranged for interviews. Verbal consent was obtained from all participants which was witnessed and formally recorded. Capacity was assessed by verbally discussing the participant information sheet with participants, giving them the opportunity to ask questions and asking them to sum-marise their understanding of the study.

## Interview procedures

Interviews were based on a semi-structured interview guide (S1 Appendix) drafted following a literature review by the first author in consultation with other members of the research team and two Lived Experience Researchers. Questions related to participants' experience of accessing and receiving mental health care and their priorities for improvement. The interview guide was pilot tested through a practice interview conducted by the first author with Nafiso Ahmed (a co-author and research fellow on the team, who is also the parent of an autistic child). This pilot interview helped assess the clarity, flow and length of the questions. Minor adjustments were made to wording and the addition of prompts to improve participant comprehension and encourage elaboration. The topic guide was further reviewed following the first two interviews to ensure that the questions were clear, acceptable and relevant to the objectives. Participants were told they could take a break or terminate the interview at any time and were provided with links to support services. The last author, an experienced consultant psychiatrist with extensive expertise in research and mental health service delivery, provided clinical oversight and was available for consultation if any risk or safeguarding concerns arose. At the end of the interviews, participants were provided with a debriefing sheet (S2 Appendix) which included links to support services such as Samaritans.

## Data collection and management

Data were collected between 26th June and 21st July 2023. The interviews were conducted by the first author, who received comprehensive training in qualitative interview techniques as part of their MSc course and from the study team.

All 13 interviews were conducted remotely via Microsoft Teams online video calls (no telephone or in-person interviews were used). Interviews lasted between 40 and 50 minutes, were audio recorded and then transcribed verbatim using a third-party transcription service approved for use by UCL. Any potentially identifiable information in the transcripts was removed and replaced with pseudonyms.

## Analysis

The transcribed and anonymised data were analysed using reflexive thematic analysis [25], whereby the transcriptions were carefully and systematically coded by the first author and then developed into common, underlying themes. Firstly, the first author read and re-read the transcripts, and made notes of initial ideas or points of interest in relation to the dataset as well as individual data items. Next, transcripts were imported into NVivo Pro V12 and the data were systematically coded, with code labels attached to segments of potentially interesting, relevant or meaningful data. In the third stage, abstract themes based on the assigned codes were constructed based around the data and the researcher's own knowledge and insights. At this point, the transcripts were sent to two separate researchers: Nafiso Ahmed, a research fellow at UCL, and Farisa Dar, an MSc student at UCL. Both researchers independently generated codes for one and two transcripts respectively. Next, the themes were reviewed and developed, and a hierarchical structure established. At this stage, a draft thematic table was shared with an autistic Lived Experience Researcher. She provided detailed written feedback confirming their relevance and suggesting minor refinements to better reflect autistic experiences such as consideration of communication barriers in accessing services and the role of special interests. This feedback informed refinements to theme naming and framing to ensure the final themes better reflected autistic lived experience.

## Reflexivity

The researchers acknowledged their own potential biases and preconceptions throughout the research process. Reflexivity was integrated into the analysis and researchers reflected on their own backgrounds and experiences that might have influenced the interpretations and findings. As the researcher who conducted the interviews and led analysis of all transcripts, I am a white British former teacher with experience working one-to-one with autistic people (many of whom had mental health conditions) in my previous role as a tutor with Mencap. I also have significant experience working with autistic children and their parents in a school setting. Although my personal experience with autistic adults and children was useful, I was careful to remind myself that it could also affect my objectivity and influence my emotional response to the information the participants were sharing with me as an interviewer.

Other members of the team included Nafiso Ahmed, who independently coded one transcript; she is a research fellow at UCL with clinical experience of working in mental health services and is a black woman and parent of an autistic child. Farisa Dar, an MSc student who independently coded two transcripts, is a Pakistani woman with experience working with autistic children at a children's therapy centre. Other authors, Tamara Pemovska, Sonia Johnson and Bryn Lloyd-Evans reviewed and commented on the draft themes. Sonia Johnson and Bryn Lloyd-Evans bring experience of working as practitioners in mental health services as well as of conducting research into potential service adaptations for autistic people. Additionally, the involvement of two Lived Experience researchers, who are autistic adults, was crucial in ensuring that the research remained grounded in the realities of those directly affected by the topic.

Our research was informed by a neurodiversity-affirming perspective where autism is viewed as natural human variation rather than a deficit. This perspective likely shaped our focus on participants' lived experiences, while giving less emphasis to biomedical or deficit-based framings of autism.

 

## Results

Thirteen autistic adults with experience of accessing and receiving mental health care in the UK volunteered and participated in the study. The self-identified gender, age, ethnicity, sexual orientation, use of mental health services and whether they have a formal autism diagnosis are shown in Table 1.

Thematic analysis of the interview transcripts identified four themes which are listed in Table 2 along with their sub-themes.

**Table 1. Participant characteristics (n = 13).**

| Characteristic | n (%) |
| --- | --- |
| Gender | |
| Male | 5 (38) |
| Female | 6 (46) |
| Non-binary | 2 (15) |
| Age | |
| 18-24 | 3 (23) |
| 25-34 | 6 (46) |
| 35-44 | 3 (23) |
| 45-54 | 1 (8) |
| Ethnicity | |
| Bangladeshi | 1 (8) |
| Chinese | 1 (8) |
| Indian | 1 (8) |
| White and Black African | 1 (8) |
| White and Asian | 1 (8) |
| White British | 8 (60) |
| Sexual orientation | |
| Asexual | 1 (8) |
| Bisexual | 5 (38) |
| Heterosexual | 4 (31) |
| Queer | 2 (15) |
| Prefer not to say | 1 (8) |
| Type of mental health service used | |
| Early intervention for psychosis service | 4 (31) |
| Adult community mental health service | 11 (85) |
| Crisis resolution and home treatment | 5 (38) |
| Psychiatric inpatient service | 8 (60) |
| Children and adolescent mental health services | 3 (23) |
| Time spent receiving mental health support | |
| 1-5 months | 1 (8) |
| 6-12 months | 0 |
| 1-5 years | 3 (23) |
| Over 5 years | 9 (69) |
| Autism identity | |
| Formally diagnosed | 12 (92) |
| Self-diagnosed or suspect I am autistic | 1 (8) |

**Table 2. List of themes and subthemes.**

| Theme | Subtheme |
|---|---|
| 1. Lack of knowledge about autism among mental health professionals | i) Stereotyped/outdated views of autism<br>ii) Impact of masking on professional understanding of autism<br>iii) Unsuitable mental health treatment for autistic people<br>iv) Disbelief of autistic identity by mental health professionals |
| 2. Unmet needs of autistic adults receiving mental health care | i) Practical issues around accessing care<br>ii) Sensory needs<br>iii) Lack of person-centred care<br>iv) Failure to consider autism alongside mental health condition(s)<br>v) Dismissed due to autism<br>vi) Failure of mental health professionals to listen to opinions of autistic adults |
| 3. Lack of consistency in mental health support for autistic adults | i) The importance of connection and continuity with familiar people in mental health care<br>ii) The importance of consistent appointments and timeliness |
| 4. Priorities for improvement | i) Adaptability and person-centred care<br>ii) Consideration of autism alongside mental health condition(s)<br>iii) Understanding of autism<br>iv) Simple communication adjustments for productive interactions<br>v) Early diagnosis of autism<br>vi) Practical considerations |

## 1. Lack of knowledge about autism among mental health professionals

Many participants felt practitioners held stereotyped and outdated views of autism, compounded by a failure to understand masking (where autistic people suppress or modify their autistic behaviours to align with dominant social norms), making it more difficult for participants to be heard or understood and impacting practitioners' ability to provide effective mental health support:

### i) Stereotyped/Outdated views of autism

Several participants reported stereotyped and outdated ideas about autism: "[Mental health professionals] understand only a very black and white, stereotypical sense. Little understanding beyond the person in the street". (P3)

Participants felt that professionals' understanding was often limited to widely known traits such as difficulty with eye contact. This prevented them from considering other aspects of autism and led to one participant being dismissed as *not* autistic:

> "I mentioned to my psychiatrist 'Oh, I think I might be autistic; can I be assessed?' He was like, 'Well, you're not autistic, you're sat here talking to me and giving me eye contact.'" (P1)

Multiple female participants regarded this stereotyped view of autism as particularly problematic for autistic girls and women, and one participant observed that outdated ideas associating autism with males, and conditions like borderline personality disorder (BPD) with females, could result in misdiagnosis:

> "I've spoken to a lot of girls my age and older who haven't been diagnosed [with autism] until their late 20s, 30s, who were misdiagnosed... BPD seems to be a massive one that people are misdiagnosed with... I think I know of about maybe 10 girls my sort of age who were diagnosed with BPD and then diagnosed with autism years later." (P4)

### ii) Impact of masking on professional understanding of autism

The issue of masking compounds the problem of mental health professionals misunderstanding the nature of autism. Many participants reported that they regularly mask their autism, impacting professionals' understanding and delaying appropriate support:

> "I tend to mask more [when I'm stressed]. So, sometimes I'll pick up the phone, like in substantial crisis, and I'll be like, [cheerily] 'Hi, I'm really struggling just now and I need help badly, is there anything you can do to support me?' And […] I've been told, 'You sound fine, why are you on this phoneline?'" (P8)

### iii) Unsuitable mental health treatment for autistic adults

Many participants reported that clinicians' lack of knowledge of autism led to treatment they found unsuitable or patronising and offered generic solutions such as mindfulness and practising good sleep hygiene:

> "I found the treatment […] very patronising […] it was like breathing exercises and stuff […] I was saying [to a psychologist], 'I already know about breathing practices – here's why they don't work for me.' 'Oh, thank you for sharing that with me. Let's do the breathing practice.'" (P6)

This is reflective of a mismatch between standard mental health interventions and the unique needs of autistic adults. The same participant went on to explain how cognitive overload and sensory issues – common experiences for autistic people – made such techniques ineffective or distressing:

> "A lot of the anxiety is, I've got too much going on in my brain at any given moment to actually remember anything and I won't be able to remember doing a breathing exercise anyway, and also it's making me feel like I'm choking and […] all this stuff is exhausting, it's tiring for me to do." (P6)

### iv) Disbelief of autistic identity by mental health professionals

**"'You, really? You don't look autistic,'... those sort of comments just aren't helpful." (P1)**

Instances of disbelief from mental health professionals often stemmed from stereotyped views and a lack of understanding regarding the ways autism manifests.

One participant described how, after adaptations had been made to accommodate some of her needs, she overheard a psychiatrist in conversation with colleagues suggesting that the adaptations were unnecessary:

> "I don't think she really needs all these things." […] I had a healthcare assistant who [would] tell me that a lot of staff had said in meetings that they didn't feel that I needed all the adaptations that I asked for. Maybe it's because I'm a girl, maybe it's because I've achieved okay academically, they just assume that my life is so easy, or that I'm just playing up to it." (P10)

### 2. Unmet needs of autistic adults receiving mental health care

Many participants described how a failure to take account of needs related to their autism alongside their mental health condition(s) impacted their treatment in various ways. Participants experienced practical issues around accessing care,

lack of consideration of their sensory needs, lack of person-centred care, lack of consideration of autism alongside their mental health needs and feelings of being dismissed from services because of their autism.

### i)   Practical issues around accessing care

Several participants described challenges in *accessing* suitable care; these challenges often arose from difficulties with communication methods such as phone calls, a common aversion for autistic people.

> "I'm not very confident because a lot of it is very appointment-based and I'm not very good- like phone calls, I just hate phone calls. Like even Mum still calls the doctor's to book my appointments and stuff. It makes me feel very kind of awkward" (P11)

General difficulties navigating the mental health care pathway were also highlighted, with participants having to speak to several people during the initial assessment process:

> "The first GP told me, 'If you're not losing touch with reality you don't have bipolar.' The next GP was like, 'Well, it's not right that you're like feeling so down. You're young, you should be going to parties and stuff. So, we'll put you on antide-pressants.' That made me kind of manic… And then I eventually got a referral to the CMHT." (P9)

This lack of continuity was identified by another participant as particularly problematic for autistic people, compounding their difficulties in accessing the right care:

> "That's not uncommon for autistic people, to find sudden change, and like new people and stuff really overwhelming." (P1)

### ii)   Sensory needs

Many participants described difficulties related to sensory overload within mental health settings, such as noise, bright lights or crowded and chaotic environments, and the failure of services to address this:

> "As an autistic person [the outpatient clinic is] quite overwhelming, in that like busy waiting area, bright clinical lights. I suffer quite a lot with sensory difficulties and get quite overwhelmed in busy places." (P1)

### iii)  Lack of person-centred care

**"I am not a cat, I do not fit in a box (P7)"**

Several participants expressed frustration with the lack of individualised treatment plans which addressed the wide range of autistic s traits and their own specific issues and concerns and instead imposed a *one-size-fits-all* approach, making them feel like *tick box* cases:

> "I feel like they don't listen, they just make assumptions about you […] and you're just treated like a tick box case. […] they don't try to make treatment something that's very individualised." (P10)

Participants wanted recognition of the diversity of autistic experiences and treatment to be tailored to their specific needs, preferences and communication styles. This included active involvement in decisions about their care and having their perspectives genuinely considered.

iv)  **Lack of consideration of autism alongside mental health condition(s)**

Some participants described experiences consistent with diagnostic overshadowing, where services failed to adequately consider the interplay between their autism and co-occurring mental health conditions, resulting in ineffective and impersonal interventions.

> "It's either, you get seen as kind of an autistic person and that's the primary thing and everything stems from that […] or you get treated as like a mentally ill person and everything else stems from that." (P6)

v)  **Dismissed due to autism**

> **"'You're too complicated for us'." (P7)**

Some participants said they were dismissed from treatment *because* of their autism:

> "'It's not our speciality, we don't deal with autistic people, we deal with people people,'" (P5).

According to one participant, fear of being dismissed from services led them to expend more energy on masking.

> "[I use] a lot more of my energy to sort of present as humanly as possible to people, so that they don't automatically just sort of stick the label on me." (P8).

One felt that her autism needs were not only overlooked but meant she received *less* support related to her BPD than her neurotypical peers:

> "Instead of giving you more help, we're going to give you less help. So with people with personality disorder [without co-occurring autism] they would say, 'Oh well we can give you intervention and it will slowly, slowly seep in. But actually, you're autistic, that's even harder, you're less likely to be able to change, therefore we can't help you.'" (P4)

vi)  **Failure of mental health professionals to listen to the opinions of autistic adults**

A common desire expressed by participants to have their own views respected and taken seriously by mental health professionals, showed their frustration with not being heard or understood:

> "I wish our opinions about ourselves were taken more seriously." (P8)

3.  **Lack of consistency in mental health support for autistic people**

Most participants expressed concerns about a lack of consistency within mental health services; factors such as changing appointments, encountering different, unfamiliar professionals and experiencing unpredictable shifts in support all contribute to heightened anxiety and hinder the benefits of various treatments.

"Inconsistency is something that I find really difficult, and I kept having different therapists, and they would like chop and change the days of my appointments with like no notice. I don't cope well with that, and I think that's not uncommon for autistic people." (P1)

i) **The importance of connection and continuity with familiar people in mental health care**

Many participants expressed concerns about services where they were offered support by ever-changing unfamiliar professionals and described how this hindered the development of a strong therapeutic alliance and thus the success of their treatment:

"The whole model of the crisis team just didn't work for me because I couldn't cope with it being different people coming to see me every day." (P5)

One participant spoke of the significantly positive impact of seeing the same professional over time.

"Seeing the same person each week for two years. She got to know me […] She knew my oddisms and my quirks and could deal with them […] I could be open and honest with her. And I would say probably that's the best care I've received."

ii) **The importance of consistent appointments and timeliness**

Consistency around location, time and regularity of appointments was particularly important for several participants. Many participants recounted times when mental health practitioners arrived late or at unexpected times and the negative impact this had on wellbeing and trust:

"They just don't seem to get things like my issues with routine… They're not getting why I'm grumpy because they've turned up an hour late." (P7)

This is particularly relevant for autistic people, many of whom depend on routine and predictability to manage anxiety and sensory sensitivity.

4. **Priorities for improvement**

Participants shared a range of priorities for enhancing mental health services for autistic adults. These suggestions encompassed adaptability, improved understanding and consideration of autism among mental health professionals, effective communication strategies, and considerations for earlier diagnosis. Throughout many of the interviews an underlying theme of co-production emerged, with the development of training, intervention adaptations, and tailored treatment plans being developed with the active participation of autistic people.

i) **Adaptability and person-centred care**

Participants emphasised the need for services to be adaptable, tailored and responsive to the unique needs and presentations of autistic people. This included personalised treatment plans that account for individual differences:

"Not everyone's going to find the same things useful. I think just being able to ask patients like, 'Okay, what works for you? What doesn't work for you?'" (P1)

## ii) Greater consideration of autism alongside mental health condition(s)

Many participants emphasised that consideration of autism alongside mental health presentations should be improved:

"The priority ought to be making sure that the interventions used for autistic patients are actually suitable for autism specifically as well as whatever else is going on." (P6)

One participant described how consideration of her individuality would improve her experience:

"I actually talk a lot when people ask me direct questions, or when people are talking about my special interests. So you know, get to know me as a person because that's really nice." (P10)

One participant felt that consideration of her autism should be balanced with her individual identity: "I wouldn't mind if they know my [autism] but I'm not just [autism], if that makes sense." (P4)

Another explained how consideration of her autism alongside her mental health condition would lead to more effective and holistic interventions: "[Consideration of my autism as well as my eating disorder would have definitely changed my experience of care] because my autism plays a really big part in my eating disorder." (P10)

## iii) Improved knowledge of autism among mental health professionals

All participants felt that mental health professionals needed further training to understand their autistic patients, with some saying that this could only happen with considerable input from autistic people themselves: "Mental health professionals really need to be trained in autism. So some type of accredited training […] developed by people with autism […] there has to be a real co-production." (P2)

Several participants described the benefits of working with mental health professionals who had knowledge of autism, with one participant highlighting how it enabled him to stop masking.

'What made her amazing was, as soon as she came into the living room… the tone she set was, "Where would you like me to sit? Is there anything that you don't want me to touch or anything like that?" […] Just by her saying that, acknowledging that […] it just showed that she understands my condition, and she's respectful […] I just felt completely comfortable around her, and I used to stim in front of her.' (P2)

## iv) Simple communication adjustments for productive interactions

Several participants suggested simple communication adjustments to make interactions more productive such as not expecting eye contact. One participant described how a professional simplified questions and allowed advance preparation, avoiding the type of open questioning that many autistic people find difficult:

"A speech and language therapist gave me these forms where, before each meeting […] it would have, 'What's gone well this week, what's not gone well this week, is there anything I want to ask about?' […] It kind of broke it down so there isn't that just, 'Right, so how are you doing?' kind of big open question." (P5)

Clear and direct communication from mental health professionals in general was felt to be a priority, supporting difficulties associated with unpredictability:

"Being able to communicate as transparently as possible. Saying, 'This is exactly what we can offer you.' The options of, 'Which do you think would be the best for you?' As opposed to, 'We're going to decide what we think's the best for you.'" (P8)

### v) Earlier diagnosis of autism

Earlier diagnosis of autism was identified as key for improved treatment outcomes and was considered especially important for autistic females: *"Probably more investment in […] diagnosis for children because I think with females, they don't even start to recognise it until a certain age."* (P11)

One female participant spoke about how her misdiagnosis of BPD was not recognised until she had an autism diagnosis:

"I got diagnosed with autism last year they said, 'Yeah, you absolutely don't have borderline personality disorder.' Like *where's that come from?* And I just think, like in my experience at least, mental health services need to be more aware of what autism can look and present like." (P1)

### vi) Awareness of sensory needs

Several participants made relatively simple suggestions as to how their sensory needs might be addressed in clinical settings which were described as often unpredictable environments Table 3:

"I mean a small adjustment might even be having ear defenders for patients to use […] like 'Are you finding the noise tough? We might have something that can reduce the noise.'" (P9)

Other participants shared positive experiences when their sensory needs were considered:

"My local [psychiatric unit], […] has a sensory room, which is awesome. They've got lights, so they're really cool, I love them." (P7)

**Table 3. Participant priorities for improving mental health services for autistic adults.**

| Priority | Key suggestions | Illustrative quote |
| --- | --- | --- |
| Adaptability and person-centred care | Flexible, individually tailored treatment plans; ask patients what works for them | "Not everyone's going to find the same things useful… what works for you? What doesn't?" (P1) |
| Greater consideration of autism alongside mental health conditions | Treat autism and mental health conditions as intertwined rather than one overshadowing the other | "My autism plays a really big part in my eating disorder." (P10) |
| Improved knowledge of autism among professionals | Mandatory autism training co-produce with autistic people | "Mental health professionals really need to be trained in autism… developed by people with autism… real co-production." (P2) |
| Simple communication adjustments | Direct questions, avoid open-ended questions, allow preparation time, no forced eye contact | "It broke it down so there isn't that just 'Right, so how are you doing?' kind of big open question." (P5) |
| Earlier diagnosis of autism | Increased investment in timely diagnosis (especially for females) | "More investment in… diagnosis for children because with females they don't even start to recognise it until a certain age." (P11) |
| Awareness of sensory needs | Offer ear defenders, quiet spaces, sensory rooms or simple enquiries about sensory triggers | "A small adjustment might even be having ear defenders… 'Are you finding the noise tough?'" (P9) |

## Discussion

In interviews with thirteen autistic adults with experience of mental health services in England, participants consistently described widespread failures to adequately recognise and address their unique needs within existing services. All highlighted the vital role of professionals' understanding of autism for effective support, yet many felt that this essential knowledge was regularly lacking. The absence of recognition for their autism-specific needs was also a common experience leading to feelings of neglect, frustration and inadequate treatment. Inconsistency within services regarding varying appointments and unfamiliar professionals was highlighted by the majority, resulting in heightened anxiety and hindered treatment benefits.

Adaptability emerged as a key focus for improvement, with many participants saying that one-size-fits-all approaches are ineffective. They stressed the need for improved understanding and knowledge when addressing their specific needs, along with more effective communication and strategies and earlier diagnosis (particularly among female participants).

This study is one of the few qualitative studies exploring autistic people's own experiences of mental health services in general and is unique in directly seeking suggestions for improvements from the participants themselves. Exploring autistic people's own views provided a comprehensive understanding of the particular challenges they face within mental health services and the general focus provided a holistic insight into overarching patterns, systemic issues and potential areas for improvement. Interviews provided a more flexible approach than previous studies, enabling the interviewer to adapt to each participant's individual needs and communication styles and provide clarification when necessary. This helped identify a wide range of relatively simple and systemic adaptations to improve mental health care for autistic people in England from the perspective of autistic people themselves.

## Findings in context

These experiences are generally consistent with previous research which demonstrated how services frequently overlook the interplay between autism and mental health; one systematic review of qualitative studies, for example, found that not only do mental health services fail to support autistic adults, they can even cause them harm [12]. Likewise, the feeling expressed by several of our participants of being dismissed by specific services for being *too complicated* due to their autism, was an experience reported in previous research where individuals felt rejected by services ill-equipped to handle the complexity of autism [12]. Our study aimed not only to investigate the impact of these general findings upon autistic adults as individuals but also to explore strategies for service improvement proposed by autistic participants.

One important issue identified in our study is the persistent shortage of professionals proficient in both autism and mental health. This scarcity has been previously noted [26] and shown to be a significant barrier to appropriate mental health support [7,8]. Other qualitative studies have also reported that this lack of expertise led to patients' needs being overlooked, exacerbating feelings of anxiety and exhaustion [27,28].

The phenomenon of masking and its tendency to amplify mental health challenges has been previously reported, with links identified between masking and heightened anxiety, depression and suicidality [29,30]. Lack of awareness of masking amongst mental health professionals often compounds the disconnect between patients and clinicians [31].

Our study also confirms that communication barriers can impede effective care, with autistic people encountering difficulties conveying their needs, while mental health professionals may struggle to understand these needs. This finding is consistent with the 'double empathy problem', or mutual misunderstanding between people with very different experiences of the world [32], specifically between autistic and non-autistic people due to differences in communication styles, sensory experiences and ways of interpreting the world. Healthcare professionals can help address the double empathy problem by adopting clear and direct communication, allowing extra processing time and actively seeking clarification rather than assuming understanding.

Despite these challenges, our research offers promising paths for improvements, with participants emphasising the value of tailored interventions and coproduction of care. This positive aspect aligns with prior research advocating for increased autism training for professionals [6,8,19]. However, one crucial caveat is evident in our study which sets it apart from much previous research and signals a shift towards more inclusive and effective care models: our participants stressed the vital importance of involving autistic people themselves in the design and implementation of such training.

Similar challenges to those reported in mental health services have also been documented across other areas of healthcare for autistic adults. A recent scoping review of experiences and perceptions of physical healthcare among autistic patients identified common barriers including communication difficulties, sensory sensitivities, inflexible systems and lack of clinical knowledge of autism, often resulting in unmet needs, negative perceptions of care and avoidance of services [33]. These similarities suggest systemic issues affect autistic adults across healthcare domains which reinforces the urgency of co-produced improvements beyond mental health services alone.

## Strengths and limitations

A key strength of this study is the involvement of autistic people throughout the research process, ensuring interview materials and themes were relevant. Recruitment was conducted online via the National Survivor User Network's weekly bulletin, which reaches over 5,000 individuals with lived experience of mental ill-health. Our sample had more female than male participants, challenging the underrepresentation of autistic females in research [34]. It was also diverse in terms of sexual orientation and gender identity, reflecting the higher prevalence of non-heterosexuality and gender non-conformity in autistic populations [35]. This study is among the few qualitative investigations into autistic adults' experiences with mental health services and is unique in gathering suggestions for improvement directly from participants. Their insights offered a comprehensive understanding of systemic challenges and potential solutions. Using interviews rather than surveys allowed for flexibility and adaptation to individual needs, helping identify a broad range of simple and systemic changes to enhance mental health care for autistic adults in the UK.

However, there are limitations to the study. The topic guide and recruitment materials targeted autistic adults without intellectual disabilities, limiting relevance for those with such co-occurrences. While the study information and data collection materials were designed to be accessible, someone with a significant intellectual disability would be unlikely to be able to give informed consent on the basis of them alone. The study also did not assess or differentiate participants according to levels of support needs, an important source of heterogeneity within autistic populations that may influence experiences of accessing and receiving mental health care. Recruiting exclusively through NSUN enabled direct access to autistic adults with lived experience of mental health difficulties, avoided clinician gatekeeping and generated a diverse sample. However, recruiting from this single source may have excluded individuals disengaged from service user networks, possibly skewing the sample towards more critical or reflective voices engaged in lived-experience advocacy while excluding those less connected to such communities. Lived Experience researchers did not participate in the interviews, which may have affected rapport and comfort levels. All thirteen interviews were conducted remotely via Microsoft Teams online video calls which may have excluded or disadvantaged unable or unwilling to participate via video technology, potentially limiting the range of perspectives captured. Most participants (69%) had over five years of mental health service use, so interviews focused more on care experiences than on access challenges. A relatively high proportion of the sample (60%) had experienced psychiatric inpatient services which may reflect greater representation of negative or crisis-related experiences [8]. The inclusion of one self-identified (rather than formally diagnosed) autistic participant increased the diversity of perspectives and aligned with principles that value self-identification [36]; however, self-identification carries an inherent risk that some experiences described may relate to other conditions or explanations other than autism.

## Clinical and research recommendations

Since it has been established that autistic adults experience a significantly higher incidence of mental ill-health compared to the general population [1], it is perhaps unsurprising that many of the problems identified by our participants are problems with mental health care in general. The need for individualised support based on the client's specific needs rather than generic advice is widely accepted to be good practice in mental health care generally, and there is evidence that supported self-management programmes where people develop their own coping strategies can be highly effective for both autistic and non-autistic people [37]. Research has also indicated the usefulness of trauma-informed approaches and validation of people's experience [38] and the importance of continuity of care [39] in all areas of mental health care. Better mental health care in general would, in other words, result in better mental health care for autistic adults.

With regard to autistic adults in particular, our findings indicate that initiatives to increase awareness such as evidence-based programmes of autism training for all mental health professionals designed in collaboration with autistic people may help address the substantial gaps in clinician knowledge and reduce the high rates of mental-ill health in autistic people. The lack of accessible post-diagnostic autism support for adults in the UK as highlighted in recent reviews [40], emphasises the need for mental health services to adopt a more holistic approach and to improve understanding of the specific needs of autistic people. In the short term, many of the practical considerations around access to care and sensory issues in clinical environments would be relatively easy to implement without requiring large amounts of re-training and could make a significant contribution to the experience and outcomes for autistic people accessing and receiving mental health care [19].

The study highlights the need for further research into several important areas. Firstly, as most of our interviews focused on the experiences of receiving mental health care, more research is required to explore the barriers and facilitators autistic people face in accessing mental health services. Further research is also required to understand what leads to the best outcomes and highest levels of satisfaction, potentially through quantitative analysis. Additionally, the experiences of autistic people with co-occurring intellectual disabilities in accessing and receiving mental health care requires further exploration. Research into the perspectives of carers and professionals working with autistic adults, including those with intellectual disabilities, is also necessary to develop a more comprehensive understanding of important perspectives on the challenges and opportunities in providing appropriate mental health care. Finally, it is essential that autistic people themselves play a central role in both research and the design of services addressing their mental health needs.

## Conclusions

Participants' experiences of mental health services tended to be negative, predominantly due to a lack of tailored support and understanding of autism by professionals. Positive experiences were reported when participants received carefully customised care, developed through constructive communication between themselves and mental health professionals whom they saw consistently and knew well. For mental health services to support autistic adults effectively and avoid potentially tragic consequences, it is crucial that mental health professionals receive comprehensive autism training. This training must be co-developed with autistic adults to ensure that it addresses real-world challenges. In the shorter term, simple, practical changes such as adjustments to clinical environments and more consistent care could lead to significant improvements.

More research is also required to better understand the various presentations and risk factors of mental health conditions in autistic adults including the investigation of the experiences of those with co-occurring intellectual disabilities. Further exploration of the perspectives of carers and mental health professionals working with autistic adults is also required and would likely contribute to more holistic and effective mental health care practices in general. Crucially, autistic adults themselves must be placed at the heart of both future research and mental health care, ensuring that their lived experience actively shapes the design of services addressing their mental health needs.

## Supporting information

**S1 Appendix. Interview Topic Guide.**
(DOCX)

**S2 Appendix. Interview Debriefing Sheet.**
(DOCX)

## Acknowledgments

We thank all the participants for their time and the sharing of their experiences. Many thanks also to the two lived-experience researchers, Jennie Parker and Amanda Timmerman for their feedback on the topic guide and thoughts on the emerging themes from the interviews.

## Author contributions

**Conceptualization:** Nafiso Ahmed, Tamara Pemovska, Farisa Dar, Brynmor Lloyd-Evans, Sonia Johnson.

**Data curation:** Frederick Taylor.

**Formal analysis:** Frederick Taylor, Nafiso Ahmed, Farisa Dar.

**Investigation:** Frederick Taylor.

**Methodology:** Frederick Taylor, Tamara Pemovska.

**Supervision:** Nafiso Ahmed, Tamara Pemovska, Brynmor Lloyd-Evans, Sonia Johnson.

**Validation:** Nafiso Ahmed, Farisa Dar, Sonia Johnson.

**Writing – original draft:** Frederick Taylor.

**Writing – review & editing:** Frederick Taylor, Nafiso Ahmed, Tamara Pemovska, Farisa Dar, Brynmor Lloyd-Evans, Sonia Johnson.

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
