## [Decision Letter · Decision Letter 0]

13 Nov 2025

PMEN-D-25-00324

Autistic adults’ experiences of accessing and receiving mental health care and their priorities for improvements: a qualitative study

PLOS Mental Health

Dear Dr. Ahmed,

Thank you for submitting your manuscript to PLOS Mental Health. I am sorry for the delay.  After careful consideration of the reviewer reports, we feel that your paper has merit but does not fully meet PLOS Mental Health’s publication criteria as it currently stands. Therefore, we invite you to submit a revised version of the manuscript that addresses the points raised during the review process.

Please address all of the comments raised, which you can find below and in the attached document.

We look forward to receiving your revised manuscript.

Kind regards,

Dr Karli Montague-Cardoso

Staff Editor

PLOS Mental Health

Journal Requirements:

1. Please send a completed 'Competing Interests' statement, including any COIs declared by your co-authors. If you have no competing interests to declare, please state "The authors have declared that no competing interests exist". Otherwise please declare all competing interests beginning with the statement "I have read the journal's policy and the authors of this manuscript have the following competing interests:"

2. Your current Financial Disclosure states, “This study was conducted as the dissertation project for the first author's MSc studies at University College London. Reimbursement to participants and lived experience researchers and transcription costs were paid for with funds from the UCL course budget. There was no other funding for this project.”. However, your funding information on the submission form indicates that you received funding that you did not receive funding. Please indicate by return email the full and correct funding information for your study and confirm the order in which funding contributions should appear. Please be sure to indicate whether the funders played any role in the study design, data collection and analysis, decision to publish, or preparation of the manuscript.

3. In the online submission form, you indicated that All relevant data are within the manuscript. In compliance with the protocol approved by the University College London Research Ethics Committee (Ref. 25209/001), individual transcripts cannot be publicly shared, as participants did not consent to their disclosure. Public access would also compromise patient confidentiality. Data access requests can be directed to the corresponding author.

3. Uploaded as supplementary information.

Reviewers' comments:

Reviewer's Responses to Questions

**Comments to the Author**

1. Does this manuscript meet PLOS Mental Health’s publication criteria? Is the manuscript technically sound, and do the data support the conclusions? The manuscript must describe methodologically and ethically rigorous research with conclusions that are appropriately drawn based on the data presented.

Reviewer #1: Yes

Reviewer #2: Yes

2. Has the statistical analysis been performed appropriately and rigorously?

Reviewer #1: N/A

Reviewer #2: N/A

3. Have the authors made all data underlying the findings in their manuscript fully available (please refer to the Data Availability Statement at the start of the manuscript PDF file)?

Reviewer #1: No

Reviewer #2: No

4. Is the manuscript presented in an intelligible fashion and written in standard English?

Reviewer #1: Yes

Reviewer #2: Yes

5. Review Comments to the Author

Reviewer #1: The authors have not made all data underlying the findings in their manuscript fully available for ethical reasons, which I believe is reasonable. However, it seems like thats a problem for the journal?

Reviewer #2: Thank you for the opportunity to read and review this interesting and timely manuscript on autistic adults' experiences of accessing mental health care in the UK. I provide several suggestions below which I hope will be helpful in further strengthening this manuscript.

Introduction

The writing is clear, but this section felt overly short, especially for a qualitative manuscript. Indeed, some key references to the literature are missing that struck me as relevant as soon as I read the Abstract (some examples below). Overall, I feel this section is a little under-developed.

Crane, L., Adams, F., Harper, G., Welch, J., & Pellicano, E. (2019). ‘Something needs to change’: Mental health experiences of young autistic adults in England. Autism, 23(2), 477-493.

Cooper, K., Loades, M. E., & Russell, A. (2018). Adapting psychological therapies for autism. Research in autism spectrum disorders, 45, 43-50.

Method

NIHR acronym needs explaining for non-UK readers.

'Lived Experience Researcher group' should also be explained - as this is within my field I will assume this means autistic people who are researchers and/or academics, but this needs spelling out for the reader as PLOS is a broad-interest journal.

In the Participants section, there is some discussion of the sampling strategy. But I wonder reading this whether the authors also considered the diversity of their sample (gender, age, ethnicity, SES, etc), and how this fed into their sampling and recruitment here? (I see in the Results section that a diverse sample was achieved, but this also needs discussing in the Participants section in terms of your efforts here).

'Experience of accessing and receiving mental health care' needs defining (end of P.4) - this is a huge range of services so it needs some definition. Could this be seeing a psychiatrist for medication? Attending a service for talking therapies? Other? etc.

I think I am correct in understanding that recruitment only happened via NSUN? If so, what are the implications of this for your sample and their possible experiences? I can see that the authors state they received sufficient numbers from this network, but I think this curtails the diversity of experiences you would have been able to achieve with a broader recruitment strategy (you could for example have restricted numbers recruited from expressions of interest from that network, and still gone ahead with recruiting via other networks too). This network may, for example, be more likely to have experienced trauma, etc. This needs addressing here and/or in the Discussion.

More information is needed in the Recruitment and Consent section about the information participants received - what did this tell them about the study, its aims, etc?

It would also be helpful to specify whether any specific people or groups were or would have been excluded (e.g., individuals with an Intellectual Disability), etc.

It is unclear where the interviews were conducted. If remotely, this needs to be specified, along with details of method/s and numbers using each (video call, telephone, etc).

Were participants provided with the interview questions or an idea of the topics in advance? If not, why not?

End of P.6: "implementing feedback from a Lived Experience Researcher on the relevance of the identified themes to

143 the target population" needs explaining further, with examples. Generally, references to the roles of the Lived Experience Researcher/s are lacking in sufficient detail throughout the manuscript.

Results

Good to see a relatively diverse sample.

Layout of results and table of themes is clear and as expected based on prior literature.

Themes/subthemes on P.14 relevant for a brief discussion about 'diagnostic overshadowing', which the authors may wish to include. Similarly, 'therapeutic alliance' is relevant to the themes discussed on P.16.

I think the Results section could be further improved by including either another table or an infographic of sorts to highlight the recommendations coming from your findings.

Discussion

The initial part of this section is somewhat under-developed, especially for a qualitative paper.

Your sample includes a fairly high number of people who've been psychiatric inpatients - the implications of this should be discussed.

Some discussion about the relative pros & cons of including a self-diagnosed interviewee should be discussed.

6. PLOS authors have the option to publish the peer review history of their article (what does this mean?). If published, this will include your full peer review and any attached files.

**Do you want your identity to be public for this peer review?** For information about this choice, including consent withdrawal, please see our Privacy Policy.

Reviewer #1: No

Reviewer #2: **Yes:** Dr Jade E Norris

Figure Resubmissions:

---

## [Decision Letter · Decision Letter 1]

15 Feb 2026

PMEN-D-25-00324R1

Autistic adults’ experiences of accessing and receiving mental health care and their priorities for improvements: a qualitative study

PLOS Mental Health

Dear Authors,

Thank you for your careful and thorough revisions. The manuscript has been substantially strengthened and is methodologically sound and clearly presented.

Before the manuscript can proceed further, a small number of minor clarifications are required:

1. Inclusion of a self-identified participant

In the Discussion, please revise the paragraph addressing the inclusion of one self-identified autistic participant (see reviewer comment below). The current framing in terms of “comparability” does not appear fully aligned with the aims of qualitative research and should be reconsidered. The section noting that inclusion increased diversity of perspectives and aligned with principles that value self-identification is appropriate and may be retained; however, the reference to comparability should be revised accordingly.

2. Support needs heterogeneity

Please add to the Limitations section a statement clarifying whether the study assessed or differentiated participants according to levels of support needs. Given the heterogeneity within autistic populations, differences in support needs may influence experiences of accessing mental health care and may represent an additional source of variability not examined in the present study.

3. Administrative requirements

In addition, please ensure that the following journal requirements are fully addressed:

Submission of a complete Competing Interests statement (including all co-authors).Confirmation of funding information and the order of funding contributions.Clarification of the Data Availability statement and, if applicable, a formal request for exemption due to ethical constraints.

Please provide a detailed response to the remaining reviewer comment included below and clearly indicate where changes have been made in the revised manuscript. Please also review and address the journal requirements listed below, as applicable.

These revisions are minor and are not expected to require further external review.

I look forward to receiving your revised submission.

Kind regards,

María Soledad Burrone, PhD, MPH, MD

Academic Editor

PLOS Mental Health

Journal Requirements:

Reviewer's Responses to Questions

**Comments to the Author**

1. If the authors have adequately addressed your comments raised in a previous round of review and you feel that this manuscript is now acceptable for publication, you may indicate that here to bypass the “Comments to the Author” section, enter your conflict of interest statement in the “Confidential to Editor” section, and submit your "Accept" recommendation.

Reviewer #1: All comments have been addressed

Reviewer #2: (No Response)

2. Does this manuscript meet PLOS Mental Health’s publication criteria? Is the manuscript technically sound, and do the data support the conclusions? The manuscript must describe methodologically and ethically rigorous research with conclusions that are appropriately drawn based on the data presented.

Reviewer #1: Yes

Reviewer #2: Yes

3. Has the statistical analysis been performed appropriately and rigorously?

Reviewer #1: N/A

Reviewer #2: Yes

4. Have the authors made all data underlying the findings in their manuscript fully available (please refer to the Data Availability Statement at the start of the manuscript PDF file)?

Reviewer #1: No

Reviewer #2: (No Response)

5. Is the manuscript presented in an intelligible fashion and written in standard English?

Reviewer #1: Yes

Reviewer #2: Yes

6. Review Comments to the Author

Reviewer #1: I am satisfied. Good job!

Reviewer #2: The authors have largely addressed my comments and suggestions thoroughly. For the final comment however, as the goal of qualitative research is not to compare/generalise, I'm not sure the interpretation given really works?:

"Reviewer: 17. Some discussion about the relative pros & cons of including a self-diagnosed interviewee

should be discussed.

Author response: Thank you. Added to pros and cons section of discussion (page 27,

lines 588-591): ”The inclusion of one self-identified (rather than formally diagnosed) autistic

participant increased the diversity of perspectives and aligned with principles that

value self-identification; however, it limits direct comparability with studies that

required formal diagnosis.”

Rather, the inclusion of people without a confirmed formal diagnosis does naturally lend itself to a risk that some people included have other conditions or explanations for their experiences that may not actually be autism. I think this needs amending/expanding more thoughtfully.

7. PLOS authors have the option to publish the peer review history of their article (what does this mean?). If published, this will include your full peer review and any attached files.

**Do you want your identity to be public for this peer review?** For information about this choice, including consent withdrawal, please see our Privacy Policy.

Reviewer #1: No

Reviewer #2: **Yes:** Dr Jade E Norris

Figure Resubmissions:

---

## [Editor Report · Decision Letter 2]

3 Mar 2026

Autistic adults’ experiences of accessing and receiving mental health care and their priorities for improvements: a qualitative study

PMEN-D-25-00324R2

Dear authors,

We are pleased to inform you that your manuscript 'Autistic adults’ experiences of accessing and receiving mental health care and their priorities for improvements: a qualitative study' has been provisionally accepted for publication in PLOS Mental Health.

Best regards,

María Soledad Burrone, PhD, MPH, MD

Academic Editor

PLOS Mental Health

Reviewer Comments :

In the Methods section, please revise the subheading “Measures.” Given the qualitative design and the content of this section (semi-structured interview guide and interview procedures), this terminology may be conceptually misleading. Please replace it with a more appropriate qualitative heading (e.g., “Data collection,” or “Interview procedures”).

In the manuscript text, please review the use of terminology such as “autistic symptoms.” Where appropriate, consider adopting alternative wording (e.g., “autistic characteristics” or “autistic traits”) to ensure conceptual precision and alignment with contemporary terminology preferences within autistic communities.